# Induction of a Compensatory Photosynthetic Response Mechanism in Tomato Leaves upon Short Time Feeding by the Chewing Insect *Spodoptera exigua*

**DOI:** 10.3390/insects12060562

**Published:** 2021-06-18

**Authors:** Julietta Moustaka, Nicolai Vitt Meyling, Thure Pavlo Hauser

**Affiliations:** Department of Plant and Environmental Sciences, University of Copenhagen, Thorvaldsensvej 40, 1871 Frederiksberg C, Denmark; nvm@plen.ku.dk

**Keywords:** insect herbivory, photosynthetic efficiency, compensatory process, chlorophyll fluorescence imaging, herbivory costs, non-photochemical quenching, photosystem II, singlet oxygen, *Solanum lycopersicum*

## Abstract

**Simple Summary:**

Insects such as beet armyworm *(Spodoptera exigua*) can cause extensive damage to tomato plants (*Solanum lycopersicum*). Tomato photosynthesis was clearly reduced directly at *S. exigua* feeding spots. However, neighboring zones and the rest of the leaf compensated through increased light energy use in photosystem II, possibly trigged by singlet oxygen from the feeding zone. Three hours after feeding, whole-leaf photosynthetic efficiency was as before feeding, demonstrating the compensatory ability. Thus, chlorophyll fluorescence imaging analysis could contribute to understanding the effects of herbivory on photosynthesis at a detailed spatial and temporal pattern.

**Abstract:**

In addition to direct tissue consumption, herbivory may affect other important plant processes. Here, we evaluated the effects of short-time leaf feeding by *Spodoptera exigua* larvae on the photosynthetic efficiency of tomato plants, using chlorophyll *a* fluorescence imaging analysis. After 15 min of feeding, the light used for photochemistry at photosystem II (PSII) (Φ*_PSII_*), and the regulated heat loss at PSII (Φ*_NPQ_*) decreased locally at the feeding zones, accompanied by increased non-regulated energy losses (Φ*_NO_*) that indicated increased singlet oxygen (^1^O_2_) formation. In contrast, in zones neighboring the feeding zones and in the rest of the leaf, Φ*_PSII_* increased due to a decreased Φ*_NPQ_*. This suggests that leaf areas not directly affected by herbivory compensate for the photosynthetic losses by increasing the fraction of open PSII reaction centers (q*_p_*) and the efficiency of these centers (F*v’*/F*m’*), because of decreased non-photochemical quenching (NPQ). This compensatory reaction mechanism may be signaled by singlet oxygen formed at the feeding zone. PSII functionality at the feeding zones began to balance with the rest of the leaf 3 h after feeding, in parallel with decreased compensatory responses. Thus, 3 h after feeding, PSII efficiency at the whole-leaf level was the same as before feeding, indicating that the plant managed to overcome the feeding effects with no or minor photosynthetic costs.

## 1. Introduction

Globally around 14% of agricultural production is lost due to herbivores, and the loss could be as high as 50% in the absence of insecticide application [1,2]. The damage caused by herbivores is mainly assessed as the amount of leaf tissue consumed, assuming that the leftover tissue is “undamaged”. However, this is a common misconception as the photosynthesis of the remaining tissue is also affected [2,3,4]. Photosynthesis of the remaining tissue can be suppressed by herbivory [4,5,6], but it can also be increased [7,8,9]. The photosynthetic efficiency of the remaining tissue plays an important role in how the plant will develop and overcome herbivory since photosynthesis generates the energy needed for the synthesis of compounds used in defense, such as hormones, primary and secondary defense-related metabolites, for compensatory repair and growth [10].

Tomato (*Solanum lycopersicum*) is an important crop plant worldwide, producing vegetables with rich nutritional value, but tomato plants are also susceptible to multiple pests, which can severely affect taste and nutritional value [11]. Among these, the larval stages of *Spodoptera exigua* (Hűbner), the beet armyworm, is a polyphagous insect with a wide distribution and wide variety of plant hosts, most of which are crop species such as cotton, cabbage, alfalfa, lettuce, and tomato plants [12,13,14]. Severe use of insecticides to control *S. exigua* has led to the evolution of insecticide resistance in many populations [14,15,16]. It was calculated that the economic injury level in tomato plants, e.g., the lowest population density that will cause economic damage [17], is only one *S. exigua* larva per twenty tomato plants, reflecting the severity of this pest on tomato production [18].

Plants defend themselves against herbivore attack through constitutive and inducible defenses and other response mechanisms, while herbivores, in turn, have evolved adaptations to defeat these mechanisms [19]. Diverse molecular processes regulate the interactions between plants and insect herbivores [20] and subsequent compensatory processes in the plants [7]. A better understanding of the extensive range of plant responses to herbivory can result from studies on how tissue injury changes the plant’s physiology, especially photosynthesis [21,22,23]. A wide range of leaf-level photosynthetic responses has been reported to occur after herbivory, fluctuating from increased, no change, or decreased photosynthetic impairment [22,24,25].

The way that we estimate productivity loss by herbivory in agriculture does not reflect the extent to which herbivory affects photosynthesis of the remaining leaf area [4]. By studying the photosynthetic response of plants in-depth, we can better understand how the undamaged tissue is affected. Chlorophyll fluorescence measurements have been used to explore the function of the photosynthetic apparatus and for the assessment of photosynthetic tolerance mechanisms to biotic and abiotic stresses [7,25,26,27]. Photosynthetic impairment due to biotic stresses can be readily revealed with chlorophyll fluorescence measurements due to the effects on light energy use [7,25]. Nevertheless, photosynthetic functioning is not homogeneous at the leaf surface, which makes standard chlorophyll fluorescence analysis non-representative of the photosynthetic status of the whole leaf [27,28,29]. The development of chlorophyll fluorescence imaging has overcome those challenges by allowing us to study the spatial heterogeneity of leaves [30,31,32].

Analyses of chlorophyll fluorescence imaging technique can be used to estimate quantitative photosynthetic changes in photosystem II (PSII), after herbivore attack, by evaluating the light energy used for photochemistry (Φ*_PSII_*), the energy lost in PSII as heat (Φ*_NPQ_*), and the non-regulated energy loss (Φ*_NO_*), which contributes to the formation of reactive oxygen species (ROS) as singlet oxygen (^1^O_2_) [33]. It should be noted that the heat energy loss, Φ*_NPQ_*, serves as a photoprotective process. Following the absorption of photons by the light-harvesting complexes (LHCs), the transfer of excitons to the reaction centers (RCs) and the initiation of electron transfer from PSII to photosystem I (PSI) must be well regulated to prevent “over-excitation” of the photosystems, which leads to the formation of ROS and photoinhibition [33,34,35,36]. In the light reactions of photosynthesis, at PSII and PSI, ROS, such as superoxide anion radical (O_2_^•−^), hydrogen peroxide (H_2_O_2_), and singlet oxygen (^1^O_2_) are continuously produced at basal levels that do not cause damage, as they are scavenged by antioxidant mechanisms [37,38,39,40,41]. However, under most biotic or abiotic stresses, the absorbed light energy exceeds what can be used, and thus, can damage the photosynthetic apparatus [42,43,44], with PSII being particularly unprotected [45]. The most important mechanism that protects against excess light conditions is the non-photochemical quenching (NPQ), which dissipates the over-excitation as heat within a time range from minutes to hours [46,47,48,49,50]. Non-photochemical exciton quenching is typically measured by the quenching of chlorophyll a fluorescence [7,40,51]. If this excess excitation energy is not quenched by NPQ, increased production of ROS occurs that can lead to oxidative stress [38,52,53].

Plant’s response to herbivory has evolved to minimize both damage and energy costs for defense and compensation. Since photosynthesis provides the energy for both these processes, it must also have evolved to be highly integrated with the plant’s reaction to herbivory. There seem to be a variability in the effect of feeding damage on photosynthesis, based on the feeding guild of the herbivore, but also the plant species under study and whether we measure the response on the plant level or the leaf level. Identifying the photosynthetic changes that occur in plants in response to herbivores will enable the understanding of response mechanisms to herbivore damage and identify potential mechanisms of tolerance [54]. In this study, we investigated whether (1) the photosynthetic efficiency of tomato leaves was suppressed immediately after a short time feeding by *Spodoptera exigua*, (2) if the plant shows signs of compensation in the leaf parts not damaged by herbivory, and (3) if the leaves are recovering from herbivory or maintain a suppressed photosynthetic state over time.

## 2. Materials and Methods

### 2.1. Plant Material and Growth Conditions

Seeds of tomato plants (*Solanum lycopersicum* cv. Moneymaker) were acquired from Kings seeds (Essex, UK) and surface sterilized with 70% ethanol for 1 min and 1% sodium hypochlorite solution for 10 min, followed by 6 washes with sterilized water. After that, they were sown in 2 L pots with potting soil (clay and silica, gröna linjen, SW Horto AB, Hammenhög, Sweden), and grown in a greenhouse chamber for 8 weeks under controlled conditions, 19 ± 1/17 ± 1 °C day/night temperature, with a photoperiod of 16-h day at 180 ± 20 μmol photons m^−2^ s^−1^ light intensity, and 60 ± 5% relative humidity. Nine-week-old plants were used for the experiments.

### 2.2. Spodoptera exigua

*Spodoptera exigua* eggs were provided by Entocare (Wageningen, The Netherlands). After hatching, L1 larvae were transferred to an artificial diet (agar 28 g, cornflower 160 g, beer-yeast 50 g, wheat germs 50 g, sorbic acid 2 g, methy1-4-hydoxybenzoato 1.6 g, ascorbic acid 8 g, streptomycin 0.1 g per L) until L2 instar. Larvae were kept under control conditions at 21 ± 1 °C day-night temperature, with a 12 h light cycle and 38% ± 5% relative humidity. The L2 instar larvae used in the experiment were starved for 24h prior to exposure to tomato leaflets in order to ensure quick consumption.

### 2.3. Experimental Design

In each of the four experimental plants, the terminal leaflet of the sixth leaf was used for the experimental measurements. The leaflet was enclosed in the measurement chamber of a fluorometer (Figure 1a), and the photosynthetic efficiency was measured (“Before herbivory” measurements). One larva was added to the leaflet within the measurement chamber, and a cap was placed over to act as an enclosure (Figure 1b). After 15 min of feeding, the larva was removed, and the leaflet was measured immediately and 90 min and 180 min later (post-feeding period).

### 2.4. Chlorophyll Fluorescence Imaging Analysis

Chlorophyll *a* fluorescence was measured at room temperature (21–22 °C) using the MINI version of an imaging-PAM fluorometer (Walz, Effeltrich, Germany, https://www.walz.com, accessed on 10 June 2021), as described before [55]. Tomato leaflets were dark-adapted for 15 min before each measurement. Eight to ten areas of interest (AOI) were selected in each leaflet before herbivory (“Before”) to cover the whole leaflet area (Figure 2a). After herbivory, an AOI was added, covering each spot of herbivory (feeding spot) and one or two AOI adjacent to the feeding spot (surrounding area) (Figure 2b). Exceptions were made when the feeding spot occurred in a nearby or an existing AOI; in that case, the new AOI was added as close as possible to the feeding spot. In total, 7 AOIs were analyzed as feeding spots.

The first step of each measurement was to determine F*o* (minimum chlorophyll *a* fluorescence in the dark) with 0.5 μmol photons m^−2^ s^−1^ measuring light and F*m* (maximum chlorophyll *a* fluorescence in the dark) with a saturating pulse (SP) of 6000 μmol photons m^−2^ s^−1^. The steady-state photosynthesis F*s* was measured after 5 min illumination time before switching off the actinic light (AL). The actinic light (AL) applied to assess steady-state photosynthesis was 200 μmol photons m^−2^ s^−1^, selected to correspond with the growing light of the tomato plants. The maximum chlorophyll *a* fluorescence in the light-adapted leaf (F*m*’) was measured with SPs every 20 s for 5 min after application of the AL (200 μmol photons m^−2^ s^−1^). The minimum chlorophyll *a* fluorescence in the light-adapted leaf (F*o*’) was computed by the Imaging Win software using the approximation of Oxborough and Baker [56] as F*o*’ = F*o*/(F*v*/F*m* + F*o*/F*m*’), where F*v* (variable chlorophyll *a* fluorescence in the dark) was calculated as F*m* − F*o*. The measured chlorophyll fluorescence parameters are shown in Table 1. Representative color code images that are displayed were obtained with 200 μmol photons m^−2^ s^−1^ AL. The results of the chlorophyll fluorescence analysis are split into (a) the whole leaflet response as a mean value of all the AOIs, and (b) the response in 3 zones, feeding spots, surrounding zones, and the rest of the leaflet.

### 2.5. Statistical Analysis

Pairwise differences in chlorophyll fluorescence parameters from before to after herbivory (15, 90, and 180 min) were analyzed with Student’s *t*-test, using the IBM SPSS Statistics for Windows version 27.0, at a level of *p* < 0.05. Average fluorescence values were estimated across the AOIs for the “Before” measurements and for each of the leaf zones (“Feeding spot”, “Surrounding zone”, and “Rest of the leaflet”) directly after herbivory (15 min) and later (90 and 180 min).

## 3. Results

### 3.1. Allocation of Absorbed Light Energy at the Whole Leaflet before and after Feeding

For the estimation of the allocation of absorbed light energy before and after feeding, we measured the fraction of the absorbed light energy that is used for photochemistry (Φ*_PSII_*), the energy that is lost in PSII as heat (Φ*_NPQ_*), and the non-regulated energy loss (Φ*_NO_*), that add up to unity [50,57]. For the whole leaflet, the fraction of absorbed light energy directed to photochemistry (Φ*_PSII_*) increased from 37% before feeding to 42% directly after feeding (15 min; significantly higher than before; Figure 3a, Appendix A). Later, 90 and 180 min after feeding, Φ*_PSII_* gradually decreased to 41% and 36%, respectively (although not significantly different from before herbivory, Appendix A). In contrast, the energy fraction lost in PSII as regulated heat (Φ*_NPQ_*) decreased from 36% before feeding to 31% directly after feeding (significantly lower; Figure 3a, Appendix A). Ninety minutes after feeding, Φ*_NPQ_* decreased further to 28% but increased to 36% at 180 min, the same as before feeding (0 min). The fraction of non-regulated energy lost (Φ*_NO_*) increased from 26% before feeding to 27% and 31% directly after feeding and at 90 min after feeding, respectively (Appendix A). At 180 min, it decreased slightly again to 28% (non-significant increase and decrease, Figure 3a, Appendix A).

### 3.2. Allocation of Absorbed Light Energy at the Feeding Site, the Surrounding Zone, and at the Rest Leaflet Areas before and after Feeding

At the feeding zone, Φ*_PSII_*decreased significantly at the larval feeding spots from before to immediately after herbivory (Figure 3b, Appendix A); it increased slightly again from 90 min to 180 min but remained significantly lower compared to before herbivory. In contrast, at the surrounding leaflet zone and the rest leaflet Φ*_PSII_* was significantly higher immediately after feeding (15 min) as well as 90 min later but 180 min after feeding did not differ compared to 0 min (before herbivory, Figure 3b, Appendix A).

The energy lost in PSII as heat (Φ*_NPQ_*) at the feeding zone (Figure 4a and Figure 5) decreased significantly from before to immediately after herbivory as well as 90 min later, but 180 min after feeding it did not differ compared to before herbivory (Appendix A). In the surrounding leaflet area and the rest leaflet area, Φ*_NPQ_*decreased significantly immediately after feeding as well as 90 min later (Figure 4a and Figure 5, Appendix A). At 180 min, Φ*_NPQ_*, in the surrounding area was still significantly reduced (Appendix A) while the rest of the leaflet did not differ compared to before herbivory (Figure 4a and Figure 5).

In the feeding zones, the non-regulated energy loss (Φ*_NO_*) increased to almost double from before to after feeding (significantly higher, Appendix A), but decreased again from 15 min to 180 min (Figure 4b and Figure 5). At 180 min, it was still significantly higher than before feeding (Appendix A). Φ*_NO_* also increased significantly in the surrounding zone (Appendix A) immediately after feeding as well as 90 and 180 min after feeding compared to before herbivory (0 min). In contrast, Φ*_NO_* did not differ in the rest of the leaflet immediately after feeding as well as 90 and 180 min later.

### 3.3. Changes in Non-Photochemical Fluorescence Quenching and Electron Transport Rate before and after Feeding

The excitation energy dissipated as heat (NPQ) at the feeding zone (Figure 6a) decreased significantly (Appendix A) immediately after feeding as well as 90 and 180 min later (compared to before herbivory). At the surrounding zone and the rest of the leaflet, NPQ decreased significantly immediately after feeding as well as 90 min later. At 180 min after feeding, only the surrounding area was significantly lower, while the rest of the leaflet did not differ to 0 min (before herbivory, Figure 6a).

In the feeding zone, the electron transport rate, ETR, decreased significantly from before to directly after herbivory (15 min; Figure 6b, Appendix A) and 180 min, while at 90 min did not differ but later (180 min) was significantly lower compared to the 0 min. At the surrounding leaflet area and the rest of the leaflet, ETR was significantly higher immediately after feeding, as well as 90 min later, but 180 min after feeding did not differ compared to 0 min (before herbivory, Figure 6b, Appendix A).

### 3.4. Changes in the Fraction of Open Photosystem II Reaction Centers and Their Efficiency before and after Feeding

The efficiency of open PSII reaction centers (F*v’*/F*m’*) increased significantly directly after herbivory (15 min) and at 90 min in the rest of the leaflet (Figure 7a, Appendix A). However, it did not differ compared to other leaflet areas at all measured times (Appendix A).

The open reaction centers of PSII (q*_p_*) decreased significantly at the feeding zones from 63% to only 40% immediately after herbivory (15 min). At the same time, the fraction of open reaction centers increased to 69% in the surrounding area and to 68% at the rest of the leaflet (Figure 7b, Appendix A).

## 4. Discussion

Herbivory is an important selective pressure in most plant species, as it usually results in reduced plant fitness [7]. However, some plants are able to compensate for the resources lost to herbivory and do not suffer any reduction in growth or reproduction after a short attack [7].

Our results show that photosynthesis of tomato leaflets in response to insect herbivory show clearly differential response at the feeding zone and at the surrounding areas. While at the feeding zone, we observed a reduction in photochemical efficiency (Φ*_PSII_*), as expected, at the surrounding leaflet area, and the rest of the leaflet Φ*_PSII_* was in contrast significantly increased (Figure 3b). Thus, photosynthetic efficiency showed signs of compensation even within the same leaflet. Compensatory ability varies depending on the plant species, the amount of leaf area lost, the environmental conditions, the mode of herbivore damage, and the timing of the herbivory [7].

In contrast to the photochemical efficiency, the fraction of energy dissipated as non-regulated energy loss in PSII (Φ*_NO_*) increased drastically upon herbivory to almost double in the feeding zones, but much less so in other leaf parts. Likewise, the fraction of energy diverted into regulated heat loss (Φ*_NPQ_*) decreased much more in the feeding zones than elsewhere. Thus, our results indicate a different response to herbivory at the different leaf zones.

The increased Φ*_PSII_* immediately after feeding in the zone surrounding the herbivore damage and in the rest of the leaflet could be ascribed either to an increased fraction of open PSII reaction centers (q*_p_*) or to increased efficiency of these centers (F*v*’/F*m*’) [58]. According to our measurements, the increased Φ*_PSII_*, was due to both, as indicated by the decrease in NPQ. The NPQ parameter is primarily representing regulated thermal energy dissipation from the light-harvesting complexes (LHCs) via the zeaxanthin quencher [33,59]. In cases of an increase in the excess light energy that is dissipated as heat (NPQ), this decreases the efficiency of photochemical reactions of photosynthesis [34,40,47,49]. Accordingly, the increased Φ*_PSII_* at the zones surrounding the feeding sites and the rest of the leaflets immediately after feeding was due to the decreased NPQ that resulted in increased electron-transport rate (ETR) (Figure 6a,b). In accordance with our results, *Cucumis sativus* plants that were subject to herbivory were able to compensate for herbivore damage by increasing their photosynthetic efficiency and capacity and by using a higher proportion of the absorbed light energy for photosynthesis [7]. Compensatory photosynthesis under herbivory was explained by a higher demand on the remaining leaf area to fix larger amounts of carbon, requiring a higher proportion of the absorbed light energy for photosynthesis [7].

The simultaneous increase in the fraction of light energy that dissipates as non-regulated energy, Φ*_NO_* indicates increased ROS creation, especially at the feeding sites through ^1^O_2_ formation (Figure 4b). Φ*_NO_* involves chlorophyll fluorescence internal conversions and intersystem crossing that results in the formation of singlet oxygen (^1^O_2_) creation via the triplet state of chlorophyll (^3^chl*) [33,41,60,61]. The ^1^O_2_ formatted this way is a highly harmful ROS generated in PSII [62,63,64,65,66]. High concentrations of ^1^O_2_ can damage proteins, pigments, and lipids in the photosynthetic apparatus and trigger programmed cell death [38,42,67]. Non-photochemical quenching (NPQ) is the photoprotective mechanism that dissipates excess light energy as heat and protects photosynthesis [45,47,68,69]. Thus, the decreased NPQ (Figure 6a) resulted in increased ROS creation through ^1^O_2_ formation (Figure 4b).

The photosystem II subunit S protein, PsbS, plays an important role in triggering NPQ responses to dissipate over-excitation harmlessly, involved in the photoprotective mechanism of heat dissipation [70]. In an impaired *Arabidopsis thaliana* NPQ mutant, lacking PsbS and the violaxanthin de-epoxidase Vde1 (commonly known as *npq4 npq1*), ROS generation was enhanced [71,72], while jasmonic acid content was altered [73,74]. Jasmonic acid is an important plant hormone that regulates, among other key responses, biotic defenses [74]. In addition, the deletion of PsbS renders *A. thaliana* mutants less attractive for herbivores [71] and capable of achieving superior pathogen defense [74]. In contrast, *A. thaliana* mutants overexpressing PsbS were preferred for feeding by both a generalist (*Plutella xylostella*) and a specialist (*Spodoptera littoralis*) insect [72]. It seems that the PsbS dependent thermal dissipation may be an important adjustment between abiotic stress tolerance and biotic defense [74]. Gaining photoprotection in photosynthesis occasionally causes decreased pathogen and herbivore defense. Consequently, plants growing in environments with a high herbivory level may evolve compensatory mechanisms as a way to maximize fitness in these environments [7]. At the same time, when no herbivory occurs, a non-compensating plant may have higher fitness than a compensating one [7]. Accordingly, in environments with constantly high herbivory, the non-compensating plant would suffer reduced fitness. From this, we suggest that the decreased NPQ, especially in the feeding zones, was due to the downregulation of PsbS. Downregulation of PsbS might be a way for plants to adjust to herbivory. Moreover, the compensatory reactions in surrounding zones and the rest of the leaflets may be signaled by ^1^O_2_ formed at the feeding zone. Future research may examine the PsbS gene expression levels and/or protein levels of plant leaves after herbivore feeding.

In our experiment, while the non-photochemical quenching (NPQ) decreased immediately after feeding, it increased again in all leaf zones until the last measurement at 180 min, to a level indistinguishable from the before feeding in the rest of the leaves (Figure 6a). At the same time, Φ*_PSII_*, Φ*_NPQ_*, and Φ*_NO_* returned to before feeding levels (Figure 3a). Other studies have reported a decreased NPQ upon pathogen attack, as early as 20 min after the attack, which was attributed to a reduced amount of PsbS, and it was proposed that NPQ regulation is a fundamental component of the plant’s defense program [71]. Defense response mechanisms can be triggered by NPQ so that light energy allocation is adjusted in order to have an enhanced PSII functionality [31]. The decreased NPQ immediately after feeding was probably caused by a reduction in the protein levels of the PSII subunit protein PsbS [75]. However, during infection with virulent and avirulent pathogens, NPQ was increased, 6 to 9 h after infection [76,77], while contrasting results with both an increase and a decrease in NPQ have also been reported [78]. These opposing results are probably due to the role of PsbS protein in the NPQ process. The PsbS protein plays the role of a kinetic modulator of the energy dissipation process in the PSII light-harvesting antenna, being not the primary cause of NPQ [79]. *Arabidopsis thaliana* plants lacking PsbS (*npq4* mutant) were found to possess a process that worked on a longer timescale, taking about 1 h to reach the same level of NPQ achieved in the wild type simply in a few minutes [79].

After herbivory, all parts of the leaflets gradually reverted towards their pre-herbivory levels. The suppressed PSII functionality at the feeding zones began to balance with the rest of the leaf 180 min after feeding, in parallel with the decreased compensatory responses at the surrounding area of the feeding zone and the rest of the leaflet. Thus, 180 min after feeding, PSII efficiency at the whole-leaf level was the same as before feeding, indicating that the plant managed to overcome the 15 min feeding effects without, or with minor, photosynthetic costs.

To the best of our knowledge, the few studies using chlorophyll fluorescence imaging analysis to study herbivory have shown increased photosynthetic damage to the leaves [2,6,30,80,81] or a slight but not significant increase in the rest of the leaf area [82]. An enhancement of photosynthesis adjacent to the sites damaged by chewing herbivores maybe because the detached leaf tissue alters the amount of source tissue without affecting the amount of sink tissue, e.g., roots and stems. Thus, photosynthesis of the remaining undamaged leaf tissue that is adjacent to the damaged leaf area may increase to compensate for the demands of the sink tissues [23]. Although there are some examples of compensatory photosynthetic mechanisms in response to insect herbivore feeding [7,83], a decline in photosynthesis occurs in most cases [2,4,5,23,84,85,86,87,88]. Photosynthesis in the remaining leaves of the plants can be upregulated as a mechanism of tolerance of herbivory [54], and there are cases of compensation in photosynthesis [7,8,9] and cases of decreased photosynthetic rates [24,81]. These contradictory results can be due to different experimental strategies, including differentially time-scheduled spatiotemporal measurements [4,5,6] and/or to the lack of spatiotemporal measurements [7,8,9]. In addition, increased NPQ at the whole leaf level is considered a major component of the systemic acquired resistance in many photosynthetic species-specific responses to insect herbivory [89]. Due to global climate change, elevated average temperatures are expected to influence plant–insect interactions and increase crop damage for two reasons [90]. First, at increased temperatures, insect metabolism increases, and the accelerated insect metabolism will cause increased crop damage; and secondly, through the herbivore-induced jasmonate signaling at elevated temperatures, the plant’s ability to cool itself is blocked by reduced stomatal opening to lead to leaf overheating and reduced photosynthesis, ultimately resulting in growth inhibition [90]. Under the increased average temperatures, the NPQ reaction is an interesting topic for future research that could be species-specific.

In conclusion, our results show that photosynthetic efficiency was only locally suppressed by *Spodoptera exigua* feeding, while in other zones of the leaflets, photosynthetic efficiency increased, indicating a compensatory response within the tomato leaflets. While this increase was also obvious at the whole leaflet level, our results show that individual local zones of the leaflets react differently and that the compensatory response was strongest closest to the feeding site. For example, the drastic increase in Φ*_NO_*, the fraction of light energy dissipated as non-regulated energy at the feeding spots immediately after herbivory could not be discerned at the whole leaflet area. Thus, comparing the whole leaflet measurements to the different zones provides a better understanding of the plant’s response to herbivory. Finally, our results show a relatively fast recovery of leaves after herbivory toward the pre-feeding level.

## Figures and Tables

**Figure 1 insects-12-00562-f001:**
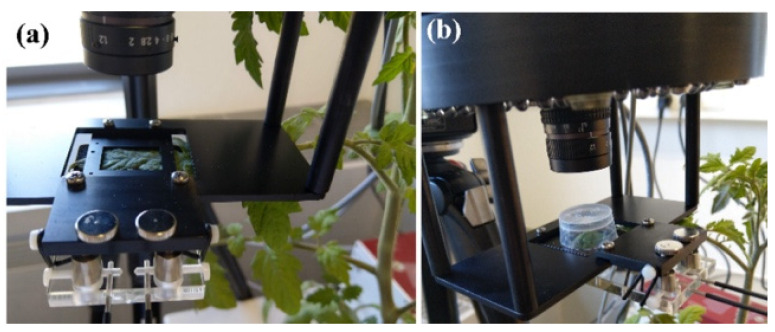
The layout of the experimental setup for chlorophyll fluorescence measurements. The terminal leaflet of the 6th leaf is placed in the instrument for the measurement before feeding (**a**), and then the larva was added on top of the leaflet (**b**). A protective cap (with holes on the top for ventilation) was placed to restrain the movement of the larva for the 15 min of feeding.

**Figure 2 insects-12-00562-f002:**
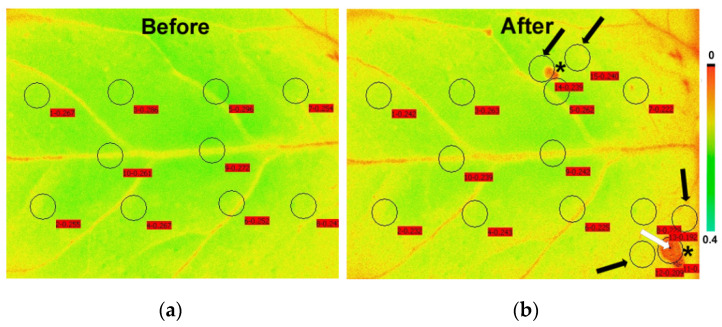
False color image of a tomato leaflet in which the color of each pixel represents the level of *F**m* (maximum chlorophyll *a* fluorescence in the dark) at the location showing the position of areas of interest (AOIs). Ten AOIs were chosen before feeding (**a**) while five additional AOIs were chosen after feeding (**b**),at the upper feeding spot (shown by an asterisk), a new AOI (shown by an arrow) was added near the existing AOIs as close as possible to the feeding spot, and another new AOI (shown by arrow) was added as a surrounding zone. At the lower feeding spot (shown also by an asterisk), one new AOI (shown by white arrow) was added in the feeding spot, and two new more AOIs (shown by black arrows) as surrounding zones. The AOIs are complemented by red labels with the F*m* value at their location. The color code on the right side shows pixel values from 0 to 0.4.

**Figure 3 insects-12-00562-f003:**
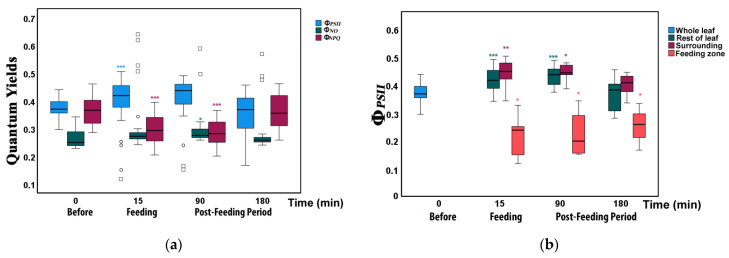
Light energy utilization in photosystem II of tomato leaflets before (0 min), immediately after insect feeding (15 min), and post-feeding period (90 and 180 min). (**a**) Allocation at the whole leaflet of absorbed light energy for photochemistry (Φ*_PSII_*, blue), regulated non-photochemical energy loss (Φ*_NPQ_*, dark green), and non-regulated energy loss (Φ*_NO_*, dark red). (**b**) The effective quantum yield of photochemistry (Φ*_PSII_*) for the whole leaflet (blue), in the feeding zone (light red), the zone surrounding the feeding zone (dark red), and in the rest of the leaflet (dark green). Boxes and whiskers indicate the tenth, twenty-fifth, fiftieth, seventy-fifth, and ninetieth percentiles. Circles and squares indicate outliers. Asterisks indicate significant pairwise differences from the Before values: *: *p* < 0.05; **: *p* < 0.01; ***: *p* < 0.001.

**Figure 4 insects-12-00562-f004:**
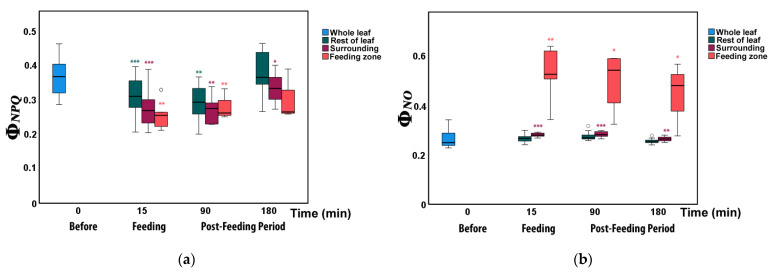
Changes in PSII quantum yield of (**a**) regulated non-photochemical energy loss (Φ*_NPQ_*) and (**b**) non-regulated energy loss (Φ*_NO_*) in different zones of tomato leaflets before (0 min), immediately after (15 min), and post-feeding period, 90 and 180 min after insect feeding. Symbols, box plots, and significances are as in Figure 3.

**Figure 5 insects-12-00562-f005:**
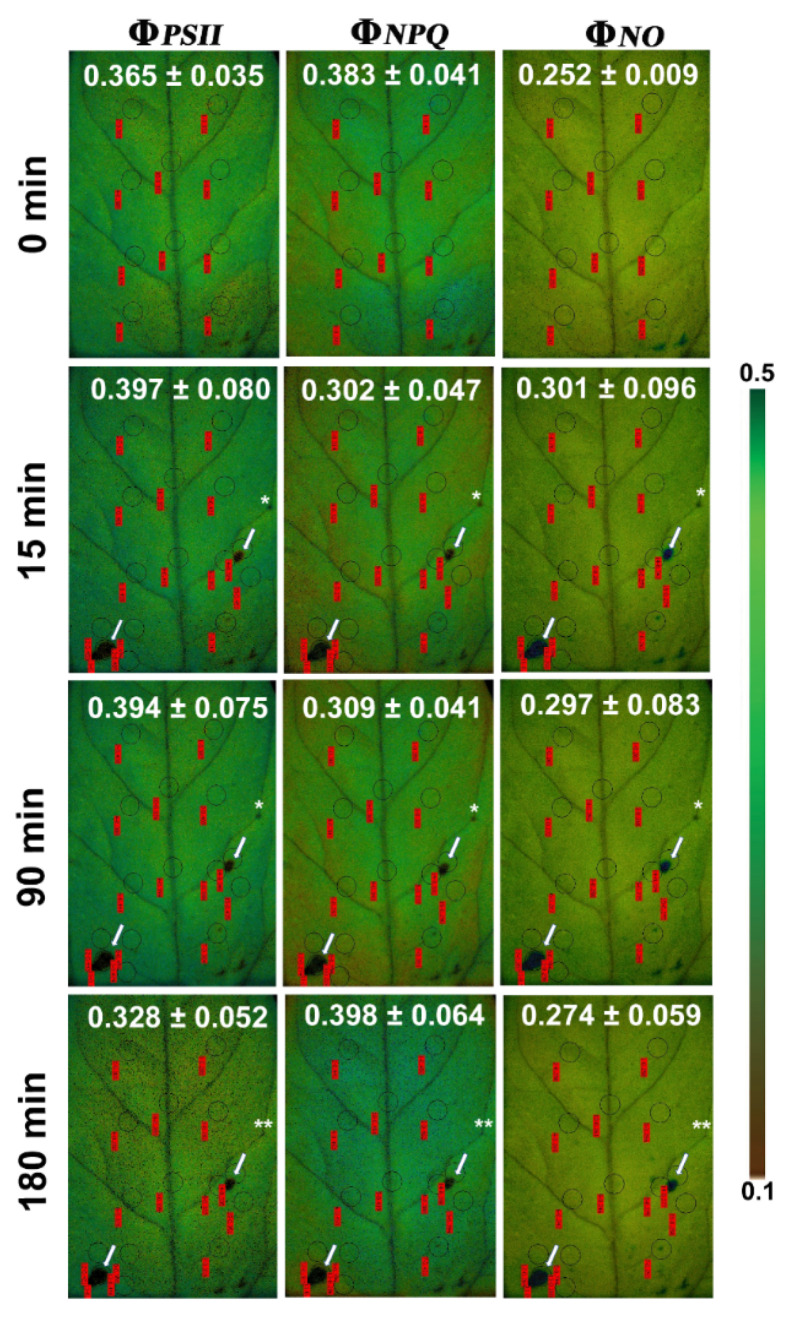
Representative color-coded images of effective quantum yield of PSII photochemistry (Φ_*PSII*_), regulated non-photochemical energy loss (Φ_*NPQ*_), and non-regulated energy loss (Φ_*NO*_) of a tomato leaflet before insect feeding (0 min; upper row), immediately after (15 min; second row), and post-feeding period (90 and 180 min; lower two rows). Ten initial measurement areas (areas of interest: AOIs) are shown in circles with their associated measurements in red labels; the corresponding values for the whole leaflet (average ± SD) are given in white. At 15 min, two AOI (shown by white arrows) were added to cover each spot of herbivory (feeding spot) and three more adjacent to and as close as possible to the two feeding spots (surrounding area). The single asterisks at 15 and 90 min mark an additional minor feeding spot, which recovered later (indicated by two asterisks at 180 min). The color code on the right side of the images shows pixel values ranging from 0.1 (dark green) to 0.5 (dark brown).

**Figure 6 insects-12-00562-f006:**
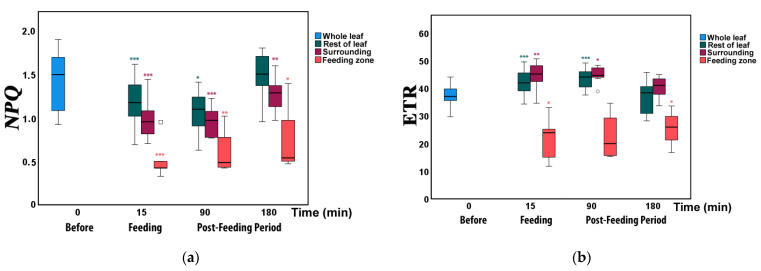
Changes in (**a**) non-photochemical quenching (NPQ) and (**b**) electron transport rate (ETR) in different zones of tomato leaflets before (0 min), immediately after (15 min), and post-feeding period, 90 and 180 min after insect feeding. Symbols, box plots, and significances are as in Figure 3.

**Figure 7 insects-12-00562-f007:**
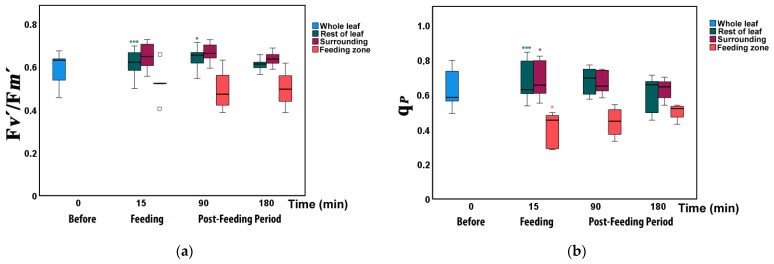
Changes in (**a**) efficiency of open PSII reaction centers (F*v*’/F*m*’) and (**b**) fraction of open PSII reaction centers (q*_p_*) in different zones of tomato leaflets before (0 min), immediately after (15 min), and post-feeding period, 90 and 180 min after insect feeding. Symbols, box plots, and significances are as in Figure 3.

**Table 1 insects-12-00562-t001:** Definitions of the chlorophyll fluorescence parameters calculated from the five main chlorophyll fluorescence parameters (F*o*, F*m*, F*o*’, F*m*’, and F*s*).

Parameter	Definition	Calculation
Φ*_PSII_*	Fraction of absorbed light energy used for PSII photochemistry	(F*m*’ − F*s*)/F*m*’
Φ*_NPQ_*	Fraction of absorbed light energy diverted into regulated heat loss in PSII	F*s*/F*m*’ − F*s*/F*m*
Φ*_NO_*	Fraction of absorbed light energy dissipated as non-regulated energy loss in PSII	F*s*/F*m*
NPQ	Non-photochemical quenching reflecting the dissipation of excitation energy as heat	(F*m* − F*m*’)/F*m*’
ETR	Electron transport rate	Φ*_PSII_* × PAR × c × abs, where PAR is the photosynthetically active radiation, c is 0.5, and abs are the total light absorption of the leaf taken as 0.84
F*v*’/F*m*’	Efficiency of open PSII reaction centers	(F*m*’ − F*o*’)/F*m*’
q*_p_*	Photochemical quenching, representing the redox state of the plastoquinone pool, or the fraction of open PSII reaction centers	(F*m*’ − F*s*)/(F*m*’ − F*o*’)

## Data Availability

The data presented in this study are available in this article.

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
