# Peer review of "Induction of a Compensatory Photosynthetic Response Mechanism in Tomato Leaves upon Short Time Feeding by the Chewing Insect Spodoptera exigua"

_insects, 2021, doi:10.3390/insects12060562_

Round 1
Reviewer 1 Report
This is the first manuscript the reviewer has seen that examines changes in photosynthesis that occur both locally and further from a site of insect herbivory. It is a worthwhile contribution to the literature. The reviewer has comments at the following lines:
Line 36 “agricultural”
Line 80 This sentence is difficult to understand. Would it be better for you to say “Measurements of chlorophyll fluorescence imaging estimate quantitative photosynthetic changes in photosystem II (PSII), after herbivore attack, by evaluating the light energy used for photochemistry (ΦPSII), the energy lost in PSII as heat (ΦNPQ), and the non-regulated energy loss (ΦNO), which contributes to formation of reactive oxygen species (ROS) as singlet oxygen (1O2) [33]. , It should be noted that the heat energy loss, ΦNPQ, serves as a protective process.”
Line 102 feeding damage
Line 324 The effect of NPQ might be affected by increased temperature, such as that induced by climate change, e.g., Havko et al. PNAS 117, 2211 (2020). It may be worthwhile to address this topic.
Author Response
Line 36 “agricultural”
We changed it accordingly.
Line 80 This sentence is difficult to understand. Would it be better for you to say “Measurements of chlorophyll fluorescence imaging estimate quantitative photosynthetic changes in photosystem II (PSII), after herbivore attack, by evaluating the light energy used for photochemistry (ΦPSII), the energy lost in PSII as heat (ΦNPQ), and the non-regulated energy loss (ΦNO), which contributes to formation of reactive oxygen species (ROS) as singlet oxygen (1O2) [33]. , It should be noted that the heat energy loss, ΦNPQ, serves as a protective process.”
Yes, it was not so clear. We change it as you suggested.
Line 102 feeding damage
Yes we changed it according to your suggestion.
Line 324 The effect of NPQ might be affected by increased temperature, such as that induced by climate change, e.g., Havko et al. PNAS 117, 2211 (2020). It may be worthwhile to address this topic.
Thank you for pointing to this nice article. We included the relevant information in the discussion at the end of the manuscript (line 394) to avoid extensive citation changes.
Reviewer 2 Report
Let me first apologize for the delay in this review. I am trying to recover from radiation damage associated with treatment for prostate cancer, and I am not the best judge of my work availability. However, I’m doing better now, so I hope you understand that my delay was unanticipated and unintentional.
From my perspective as someone who has worked on herbivory and photosynthesis most of his career, this is just a wonderful paper. The authors do an outstanding job in every aspect of their work. The introduction places this work in the context of previous research, the methods are well defined, and the results are presented in a clear and convincing fashion. Among the points I found most laudable was the discussion and how the limitations of the results were properly delineated. In the attached pdf, noted a couple of minor editorial corrections, and I offer a comment on the role of environment and quantity of injury being likely to alter or influence responses described here. However, I could have saved my effort because in subsequent paragraphs the authors make exactly the same points and expand upon these to provide the reader a balanced and nuanced perspective on their findings.
So often I find that researchers lose perspective on their results either overemphasizing their significance or understating impact and implications. I think these problems are especially common in photosynthetic research of many types, because so many biotic and abiotic factors infleunce photosynthesis. And there is, of course, the additional problem that seeing any response at a small scale is not the same as demonstrating that such a response actually influences fitness. The authors here avoid all these pitfalls, and provide a balanced, convincing article with valuable new findings. Consequently, I heartily recommend this paper for publication and look forward to seeing it 8n the literature.

Author Response
Let me first apologize for the delay in this review. I am trying to recover from radiation damage associated with treatment for prostate cancer, and I am not the best judge of my work availability. However, I’m doing better now, so I hope you understand that my delay was unanticipated and unintentional.
From my perspective as someone who has worked on herbivory and photosynthesis most of his career, this is just a wonderful paper. The authors do an outstanding job in every aspect of their work. The introduction places this work in the context of previous research, the methods are well defined, and the results are presented in a clear and convincing fashion. Among the points I found most laudable was the discussion and how the limitations of the results were properly delineated. In the attached pdf, noted a couple of minor editorial corrections, and I offer a comment on the role of environment and quantity of injury being likely to alter or influence responses described here. However, I could have saved my effort because in subsequent paragraphs the authors make exactly the same points and expand upon these to provide the reader a balanced and nuanced perspective on their findings.
So often I find that researchers lose perspective on their results either overemphasizing their significance or understating impact and implications. I think these problems are especially common in photosynthetic research of many types, because so many biotic and abiotic factors infleunce photosynthesis. And there is, of course, the additional problem that seeing any response at a small scale is not the same as demonstrating that such a response actually influences fitness. The authors here avoid all these pitfalls, and provide a balanced, convincing article with valuable new findings. Consequently, I heartily recommend this paper for publication and look forward to seeing it 8n the literature.
Never mind for the delay. Health is first of all. All other things can wait. We wish you full recovery. We also thank you for your positive comments. The minor editorial corrections were implemented.
Reviewer 3 Report
The paper described an in-door trial to evaluated the effects of short time leaf feeding by Spodoptera exigua larvae on photosynthetic efficiency of tomato, using chlorophyll a fluorescence imaging analysis.
The methods sections needs a huge improvemt and I suggest the authos to consider more replicates since only four experimental plants were used. The authors have data only from 4 leaflets. This is not representative. Therefore, the conclusion of the paper could be only for randomly reason.
Line 116 – Describe soil characteristics.
Line 119 – Nine-week-old plants corresponds to what development stage?
Line 119 – Where the study was developed?
Line 121 – S. exigua larvae came from artificial diet conditions, the authors should develop at least one generation under tomato leaves. This can influence on the insect feeding performance.
Line 124-125 – The authors said “Larvae were kept under control conditions” it was in a room? Please, specify.
Line 129 – What was the experiment design? Please, provide this information.
Line 132 – What are the dimensions of the leaflet?
Line 121-126 – The authors said “After 121 hatching, L1 larvae were transferred to an artificial diet (agar 28 g, cornflower 160 g, beer-122 yeast 50 g, wheat germs 50 g, sorbic acid 2 g, methy1-4-hydoxybenzoato 1,6 g, ascorbic 123 acid 8 g, streptomycin 0,1 g per liter) until L2-L3 instar…The L2 instar larvae used in the experiment were starved “. The sentence is kind confusing, please clarify. Why did you kept the larvae in diet until L2-L3 instar if you used only L2?
Line 133-135 – The authors have data only from 4 leaflets? This is not representative. Therefore, the conclusion of the paper could be only for randomly reason.
Line 142 – What was the room temperature?
Line 143 – The authors need to describe how the equipment work and specifications
Line 144 – Why tomato leaves were dark adapted?
Line 144 – It is not clear how the AOI were defined. Please, clarify.
Line 174 – Did the authors considered the number of AOI as replicates? If positive, this is not suitable, you oversampled the data. If note, please, clarify how many samples had you used and what you considered as sample.
Lines 185 – 187 – Perhaps it is missing something in this sentence, it is quite confusing.
Lines 292 – 296 – The authors need to explain what they mean with coherent; it is not clear and makes the sentence interpretative.
Figures – The “*” in all figures is confusing, please consider remove it or better organize.
Author Response
The paper described an in-door trial to evaluated the effects of short time leaf feeding by Spodoptera exigua larvae on photosynthetic efficiency of tomato, using chlorophyll a fluorescence imaging analysis.
The methods sections needs a huge improvement and I suggest the authors to consider more replicates since only four experimental plants were used. The authors have data only from 4 leaflets. This is not representative. Therefore, the conclusion of the paper could be only for randomly reason.
The average number of samples used in other published studies of chlorophyll fluorescence in the lab ranges from 3 to 6, which can be verified in the list of articles that follows. As we mention on line 136 “In each of four experimental plants…”, which is within this range. Our results on many (or most) of the parameters show drastic changes immediately upon herbivory. These results are highly unlikely if our results were “... only for randomly reason”, as the reviewer suggests. We therefore judge that our results are robust, but acknowledge that larger studies are needed, especially with more plant genotypes and herbivore specializations.
Plant Cell Physiol. 2011, 52, 1822–1831. (Fig. 3, n= 3). (Fig. 4, n= 3).
J Plant Physiol. 2014, 171, 23-30. (Four replicates, each consisting of an individual plant in a pot)
Plant Cell Physiol. 2016, 57, 1510-1517. (Three independent measurements)
Physiol. Plant. 2016, 158, 225–235. (Four biological replicates)
Front. Plant Sci. 2016, 7, 453 (4 replicates)
Science 2016, 354 (6314) 857-860. (Fig. 3, n = 5 biological replicates), (Fig. 4, n = 6 biological replicates)
Sci. Rep. 2017, 7, 46100. https://doi.org/10.1038/srep46100. (n= 3).
Nature Plants 2017, 3, 17033. DOI: 10.1038/nplants.2017.33 (Figure 3. n > 3).
ACS Appl. Mater. Interfaces 2018, 10, 4450−4461 (n= 5).
Plant Cell Environ. 2021, DOI: 10.1111/pce.14032 (n = 3, biologically independent samples)
J Plant Physiol. 2021, 262, 153438 (N=3, N=4)
J Plant Physiol. 2021, 260, 153404 (three replicates)
Journal of Hazardous Materials 2021, 404, 124001 (three independent biological replicates)
Molecules 2021, 26, 2984. https://doi.org/10.3390/molecules26102984 (four independent measurements)
Int. J. Mol. Sci. 2021, 22, 41. https://dx.doi.org/10.3390/ijms22010041. (n = 5).
Plants 2021, 10, 521. https://doi.org/10.3390/plants10030521 (n = 6).
Line 116 – Describe soil characteristics.
The soil used was normal potting soil: peat soil supplemented with 5% gravel (grid size: 1-3 mm), clay (grid size: 2-6 mm), limestone (pH: 5.5-6.5), special fertilizers (PG-Mix) and micronutrients (Krukväxtjord Lera & Kisel, Gröna linjen, Sweden) (lines 117-118).
Line 119 – Nine-week-old plants corresponds to what development stage?
The plants were right on the beginning of their flowering stage.
Line 119 – Where the study was developed?
The study was developed in a normal laboratory at the Department of Plant and Environmental Sciences, University of Copenhagen, Thorvaldsensvej 40, 1871, Frederiksberg C, Denmark.
Line 121 – S. exigua larvae came from artificial diet conditions, the authors should develop at least one generation under tomato leaves. This can influence on the insect feeding performance.
In this work, we are not focusing on the insect performance but using the insect to test effects on the photosynthetic efficiency of the plant. The insects are starved for 24h as mentioned which is allowing them to feed on the plant quickly.
Line 124-125 – The authors said “Larvae were kept under control conditions” it was in a room? Please, specify.
We specify the conditions in lines 126-127: at 21 ± 1 °C day-night temperature, with a 12-h light cycle and 38 ± 5% relative humidity.
Line 129 – What was the experiment design? Please, provide this information.
This information is provided in Figure 1. The plants were measured sequentially as each measurement takes around 3 hours.
Line 132 – What are the dimensions of the leaflet?
The dimensions of the leaflet were appropriate so as to fit in the chlorophyll fluorescence measurement area of 24x32mm.
Line 121-126 – The authors said “After 121 hatching, L1 larvae were transferred to an artificial diet (agar 28 g, cornflower 160 g, beer-122 yeast 50 g, wheat germs 50 g, sorbic acid 2 g, methy1-4-hydoxybenzoato 1,6 g, ascorbic 123 acid 8 g, streptomycin 0,1 g per liter) until L2-L3 instar…The L2 instar larvae used in the experiment were starved “. The sentence is kind confusing, please clarify. Why did you kept the larvae in diet until L2-L3 instar if you used only L2?
We changed the sentence to:
After hatching, L1 larvae were transferred to an artificial diet (agar 28 g, cornflower 160 g, beer-yeast 50 g, wheat germs 50 g, sorbic acid 2 g, methy1-4-hydoxybenzoato 1,6 g, ascorbic acid 8 g, streptomycin 0,1 g per liter) until L2 instar.
Line 133-135 – The authors have data only from 4 leaflets? This is not representative. Therefore, the conclusion of the paper could be only for randomly reason.
See our answer on sample sizes above.
Line 142 – What was the room temperature?
The room temperature was 21 - 22 °C.(line 144).
Line 143 – The authors need to describe how the equipment work and specifications.
This information is available at the manufactures site that is included in the manuscript. Line 144 (Walz, Effeltrich, Germany, https://www.walz.com).
Line 144 – Why tomato leaves were dark adapted?
This is done so as all the reaction centres to be open before beginning the measurements.
Line 144 – It is not clear how the AOI were defined. Please, clarify.
Lines 146-150 it is written “Eight to ten areas of interest (AOI) were selected in each leaflet before herbivory (“Before”) so as to cover the whole leaflet area (Figure 2a).” After herbivory an AOI was added covering each spot of herbivory (feeding spot) and one or two AOI adjacent to the feeding spot (surrounding area) (Figure 2b).
Line 174 – Did the authors considered the number of AOI as replicates? If positive, this is not suitable, you oversampled the data. If note, please, clarify how many samples had you used and what you considered as sample.
We have clarified the sample sizes and data treatment in the text 4 plants with seven feeding spots were analysed in total. (lines 136 and 163 ).
Lines 185 – 187 – Perhaps it is missing something in this sentence, it is quite confusing.
Yes, something was missing. The sentence was changed to: “For the estimation of the allocation of absorbed light energy before and after feeding we measured the fraction of the absorbed light energy that is used for photochemistry (ΦPSII), the energy that is lost in PSII as heat (ΦNPQ), and the non-regulated energy loss (ΦNO), that add up to unity [50,57].” (line 193-196).
Lines 292 – 296 – The authors need to explain what they mean with coherent; it is not clear and makes the sentence interpretative.
The sentence was changed to: “Thus, our results indicate a different response to herbivory at the different leaf zones”.
Figures – The “*” in all figures is confusing, please consider remove it or better organize.
Yes, you are right. The different type of asterisks was confusing. We organized better all the Figures. We replaced the asterisk of the outliers by a square.
Round 2
Reviewer 3 Report
After the incorporated suggestions I think that the manuscript is more robust and ready for been published.